# A machine learning approach for computation of cardiovascular intrinsic frequencies

Rashid Alavi[1☺], Qian Wang[2☺], Hossein Gorji[3], Niema M. Pahlevan[1,4,5]*

**1** Department of Aerospace and Mechanical Engineering, University of Southern California, Los Angeles, California, United States of America, **2** Beijing Computational Science Research Center, Beijing, China, **3** Swiss Federal Laboratories for Materials Science and Technology (EMPA), Dubendorf, Switzerland, **4** Cardiovascular Research Institute, Huntington Medical Research Institutes, Pasadena, CA, United States of America, **5** Division of Cardiovascular Medicine, Keck School of Medicine, University of Southern California, Los Angeles, California, United States of America

☺ These authors contributed equally to this work.
* pahlevan@usc.edu

**Data Availability Statement:** The datasets supporting this article have been uploaded to the Dryad Digital Repository (doi:10.5061/dryad.dfn2z353x) URL: https://datadryad.org/stash/

## Abstract

Analysis of cardiovascular waveforms provides valuable clinical information about the state of health and disease. The intrinsic frequency (IF) method is a recently introduced framework that uses a single arterial pressure waveform to extract physiologically relevant information about the cardiovascular system. The clinical usefulness and physiological accuracy of the IF method have been well-established via several preclinical and clinical studies. However, the computational complexity of the current $L_2$ optimization solver for IF calculations remains a bottleneck for practical deployment of the IF method in real-time settings. In this paper, we propose a machine learning (ML)-based methodology for determination of IF parameters from a single carotid waveform. We use a sequentially-reduced Feedforward Neural Network (FNN) model for mapping carotid waveforms to the output parameters of the IF method, thereby avoiding the non-convex $L_2$ minimization problem arising from the conventional IF approach. Our methodology also includes procedures for data pre-processing, model training, and model evaluation. In our model development, we used both clinical and synthetic waveforms. Our clinical database is composed of carotid waveforms from two different sources: the Huntington Medical Research Institutes (HMRI) iPhone Heart Study and the Framingham Heart Study (FHS). In the HMRI and FHS clinical studies, various device platforms such as piezoelectric tonometry, optical tonometry (Vivio), and an iPhone camera were used to measure arterial waveforms. Our blind clinical test shows very strong correlations between IF parameters computed from the FNN-based method and those computed from the standard $L_2$ optimization-based method (i.e., R≥0.93 and P-value ≤0.005 for each IF parameter). Our results also demonstrate that the performance of the FNN-based IF model introduced in this work is independent of measurement apparatus and of device sampling rate.

share/Z1UV0MDUmyX4nIcGEseGHAWDLT4
JslfmIDDuJ0HzIr4

**Funding:** The authors received no specific funding
for this work. This study used Framingham Heart
Study data. The Framingham Heart Study is
supported by the National Heart, Lung, and Blood
Institute (NHLBI) under contract
(HHSN268201500001I) with additional support
from other sources.

**Competing interests:** Niema M. Pahlevan holds
equity in Avicena LLC and has a consulting
agreement with Avicena LLC. This does not alter
our adherence to PLOS ONE policies on sharing
data and materials.

## 1. Introduction

General-purpose function approximators established by machine learning (ML) offer new perspectives in medical research [1]. Their accuracy, robustness and universality make them appropriate building blocks for remote health monitoring and early diagnosis [2, 3]. The possibility of developing effective ML algorithms, which could assist in the diagnosis of cardiovascular diseases, has led to a quest for reliable yet efficient classification models of cardiovascular waveforms [4–6].

Feedforward neural networks (FNNs) represent a widely-used class of neural networks (NNs) that are trained for conventional ML tasks [7–9]. These simple NN architectures include a few hidden layers connecting the input layer to the output one. While directly applying FNN models on hemodynamic waveforms (i.e., arterial pressure waveforms) seems natural and intuitive, the high dimensionality of the input signal renders naive FNN constructs as significantly limited for practical data-driven diagnosis. Therefore, it is essential to introduce an FNN architecture to extract key physiological information carried by input signals, serving as a pre-prerequisite to achieve robust and efficient classifications on arterial pressure waves. There exists a variety of reduced-order NN models due to a surging popularity in recent years that has turned the construction of such models into an active area of research in recent years [10, 11]. Common reduced-order NN approaches (e.g., Active Subspace, Proper Orthogonal Decomposition and Polynomial Chaos Expansion [12–14]) focus solely on the data and treat the underlying physical/physiological dependencies as a black box—at the expense of relying mostly on the training data. The proposed methodology of this work, on the other hand, leverages the underlying physiological information of the waveforms. Indeed, we have adopted a compressed FNN in our approach for diagnosing cardiovascular diseases based on a recently-introduced signal analysis technique called the intrinsic frequency (IF) method (briefly overviewed in what follows and described in further detail in Section 2.1).

Traditional signal analysis methods such as Fourier transform have major limitations when they are applied on waves/signals that arise from nonlinear and non-stationary systems. Time-frequency analysis methods are effective tools for analyzing such signals [15–19]. The sparse time-frequency representation (STFR) is a time-frequency method (inspired by empirical mode decompositions, or EMD [18, 19]) that can preserve intrinsic physical characteristics of a non-stationary and nonlinear wave [16, 17]. In addition, the STFR method is less sensitive to noise and can be applied to sharp signals such as those found in arterial waveforms of the cardiovascular system [20]. The mathematical formulation of an SFTR for a real signal $p(t)$ is given by $p(t) = \sum_{i=1}^{M} a_i(t) cos\, \theta_i(t)$, where $a_i(t)$ is the envelope and $\theta_i(t)$ is the phase. The time-derivative of $\omega_i(t) = d\theta_i/dt$ is called an *instantaneous frequency*. Pahlevan *et al.* [20] have applied the SFTR method on arterial pressure waveforms and have shown that these instantaneous frequencies reveal physiologically relevant information about the dynamics of the left ventricle (LV), the arterial system, and their coupling [20–26]. In particular, they have demonstrated that the instantaneous frequency oscillates around a dominant frequency ($\omega_1$) during systole (when the aortic valve is open) and subsequently switches to a different range and oscillation around a second dominant frequency ($\omega_2$) during diastole [20] (when the aortic valve is closed). These dominant frequencies are called *intrinsic frequencies* (IFs). Pahlevan *et al.* have introduced a brute-force algorithm that extracts IFs from an $L_2$ minimization problem, where the minimization is solved for all possible values of frequencies in a domain. The corresponding solution of such a formulation provides the pair of intrinsic frequencies, $\omega_1$ and $\omega_2$, that has the minimum residual in the $L_2$ optimization problem [20].

This work introduces an FNN workflow to calculate IF frequencies and other related IF parameters directly from a signal (an arbitrary pressure waveform). The FNN approach is

realized through a sequentially-reduced representation for computing IF parameters (i.e., $\omega_1$, $\omega_2$) directly from pressure waveforms. In order to train, test, and validate our model, we have employed both synthetic waveforms (reconstructed from exact IF values) and clinical waveforms. Our clinical training datasets include carotid waveforms measured by various devices (i.e., tonometry, iPhone, Vivio) from two different clinical studies (FHS [27] and the HMRI iPhone Heart Study [28]). After finalizing our model, we perform a blind clinical test by applying our model on noninvasively measured clinical carotid waveforms from 3009 patients from the FHS database.

## 2. Materials and methods

### 2.1. Intrinsic frequency method

The Intrinsic Frequency (IF) method [20] is a recently-developed systems approach to investigate the global dynamics of the cardiovascular system. As highlighted in the introduction (Section 1), the IF method utilizes a modified version of the STFR [16] in order to extract the dominant operating frequencies of the arterial blood pressure (BP) for both the coupled LV-arterial system (systolic phase) as well as the decoupled aorta after aortic valve closure (diastolic phase). In other words, the IF method models the cardiovascular dynamical system as an object rotating around an origin by considering two independent and different dynamics representing the cardiovascular system over one cardiac cycle: the coupled LV-arterial system and the decoupled LV/aorta dynamical system. In the coupled LV-aorta system, the average angular velocity of rotation (average instantaneous frequency) is defined as $\omega_1$, while the average angular velocity during diastole is defined as $\omega_2$. The two dominant frequencies $\omega_1$ and $\omega_2$ are called the first and second intrinsic frequencies, respectively. The IF method assumes that the instantaneous frequency of a coupled dynamic system is piecewise constant over time with a step that occurs at the time of decoupling (aortic valve closure). It should be noted that IF frequencies are fundamentally different than Fourier harmonics or any other resonance-type frequencies.

When applied to arterial waveforms, the IF method reveals clinically important information about the dynamics of the LV and the arterial system (as well as their interactions) in both healthy and disease conditions [21, 23, 24, 28–33]. Several clinical studies have confirmed the clinical usefulness of cardiovascular IFs in both the diagnosis and prognosis of cardiovascular diseases (CVDs) [23, 28, 30, 34]. One important advantage of the IF method for clinical applications is that the absolute magnitude of the arterial pressure wave is not required to extract the IF parameters; only the waveform morphology is needed [20, 28]. As such, IFs can be easily acquired noninvasively and inexpensively using a smartphone [24, 28], arterial applanation tonometry, or a wearable sensor. In a notable study, the left ventricle ejection fraction (LVEF) was accurately approximated by applying the IF method to the noninvasive carotid waveforms measured by an iPhone camera [28].

**2.1.1. Optimization-based IF approach ($L_2$-minimization problem for calculation of intrinsic frequencies).** For an arterial waveform $p(t)$ (e.g., of the aortic, carotid, or femoral), the IF method solves [20, 35] a nonlinear optimization problem ($L_2$-minimization) for a cardiac cycle of length $T$ through the objective function that is given by:

$$\left\| p(t) - \chi(0, T_0)[a_1 cos(\omega_1 t) + b_1 sin(\omega_1 t)] - \chi(T_0, T)[a_2 cos(\omega_2 t) + b_2 sin(\omega_2 t)] - c \right\|_2^2 \quad (1)$$

and minimized for variables $\omega_1$ and $\omega_2$, which represent the first (or systolic) and the second (or diastolic) intrinsic frequencies, respectively, and $a_1$, $b_1$, $a_2$, $b_2$, $c$, which represent the corresponding constant envelopes and intercept. Here, $\chi(a, b)$ is an indicator function defined by $\chi(\alpha, \beta) = 1$ if $\alpha \leq t \leq \beta$ and $\chi(\alpha, \beta) = 0$ otherwise. The parameter $T_0$ represents the decoupling time (or the time to closure of the aortic valve, i.e., the dicrotic notch). Such an $L_2$-

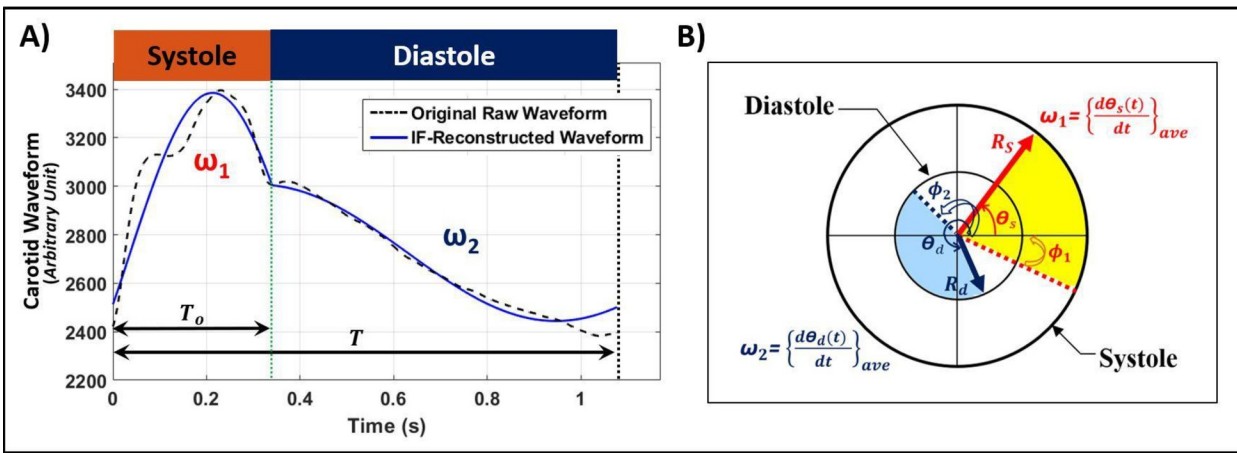

**Fig 1. Illustration of the intrinsic frequency (IF) method. A) Reconstruction of a carotid pulse waveform using the IF method.** The original raw waveform in arbitrary units (dashed black) is overlaid on a waveform (blue) that is reconstructed from the IF method. **B) IF visualization during a cardiac cycle.** The values $\omega_1$ and $\omega_2$ represent IFs during systole and diastole, respectively, and $d\theta/dt$ is the instantaneous frequency [20]. The location of the dicrotic notch is marked by the vertical green–dotted line. The values $R_s$ and $R_d$ are the envelopes of the IF reconstruction associated with $\omega_1$ and $\omega_2$, respectively (note that, in general, $R_s \neq R_d$). The values $\varphi_1$ and $\varphi_2$ are the initial intrinsic phases associated with $\omega_1$ and $\omega_2$, respectively.

minimization problem is subjected to two nonlinear constraints: continuity at the dicrotic notch ($T_0$) and periodicity of the waveform. These are formulated as

$$a_1 cos(\omega_1 T_0) + b_1 sin(\omega_1 T_0) = a_2 cos(\omega_2 T_0) + b_2 sin(\omega_2 T_0), \text{ and} \qquad (2A)$$

$$a_1 = a_2 cos(\omega_2 T) + b_2 sin(\omega_2 T), \qquad (2B)$$

respectively. After solving the non-convex minimization problem defined by Eqs (1), (2A) and (2B) for the seven optimization variables [$a_1$, $b_1$, $\omega_1$, $a_2$, $b_2$, $\omega_2$, $c$], optimum values of IF parameters are obtained. Fig 1A illustrates how the IF-reconstructed waveform represents the original pressure waveform. Further details regarding the mathematical formulation, computational procedure, scalability, convergence/accuracy, and applicability on non-invasive measurements of the IF method have been provided in previous studies [20, 28, 35–37].

**2.1.2. Reformulation of the intrinsic frequency method for the ANN-based IF approach.** Using trigonometric equations, the original formulation of the IF method (Eq (1)) can be reformulated as

$$Minimize: \; \|p(t) - \chi(0, T_0)[(R_s sin(\omega_1 t + \varphi_1)] - \chi(T_0, T)[(R_d sin(\omega_2 t + \varphi_2)] - c\|_2^2, \qquad (3)$$

where $\varphi_1$ and $\varphi_2$ are introduced as the initial intrinsic phases of the IF components associated with $\omega_1$ and $\omega_2$, respectively (see Fig 1B). The parameters $R_s$ and $R_d$ represent the two regimes of the piecewise constant envelopes for the systolic and diastolic IFs, respectively. These parameters are related to the constants $a_1$, $b_1$, $a_2$, $b_2$ of Eq (1) respectively as

$$\varphi_1 = tg^{-1}(a_1/b_1), \; \varphi_2 = tg^{-1}(a_2/b_2), \; R_s = \sqrt{a_1^2 + b_1^2}, \; R_d = \sqrt{a_2^2 + b_2^2} \qquad (4)$$

The non-dimensional ratio $R_s/R_d$ of the two envelope constants is known as the envelope ratio (*ER*) [21, 37]. The systolic IF parameters ($\omega_1$, $\varphi_1$, $R_s$) are dominated by the dynamics of the coupled LV-arterial system, whereas the diastolic IF parameters ($\omega_2$, $\varphi_2$, $R_d$) are dominated by the dynamics of just the arterial network [23, 24, 28, 30, 32, 38–41]. Fig 1B illustrates details

about the above-mentioned mathematical reformulation through a visualization of $\omega_1$, $\omega_2$, $\varphi_1$, $\varphi_2$, $R_s$ and $R_d$ during the systolic and diastolic phases.

## 2.2. Heterogeneous databases

In order to design our ANN model (i.e., train, validate, and test), we have used a mixture of carotid waveform signals from 1) clinical databases (with waveforms measured by three distinct devices: Tonometry [30], Vivio [42], and iPhone camera [28]) and 2) a synthetic database. The latter includes synthetically generated waveforms with exact IF values ($\omega_1$, $\varphi_1$, $R_s$, $\omega_2$, $\varphi_2$, $R_d$, c) in order to ensure adherence to the mathematical formulation of IF during the training process. The clinical databases enrich the training algorithm and subsequently the ANN model for eventual clinical purposes (e.g., preparation for real-world physiological variations and noise). Additionally, a portion of the clinical database is set aside before the design process to facilitate a blind external test and thereby assess the robustness/accuracy of the final model (see Section 2.7 for details).

**2.2.1 Clinical database.** Our clinical database for designing/blind-testing the model is provided by two different clinical studies: the Framingham Heart Study (FHS) and the HMRI iPhone Heart Study. Both studies include carotid artery waveforms of the participants. Since these waveforms are from three distinct devices (Tonometry for FHS [27]; Tonometry, Vivio, and an iPhone camera for HMRI [28]), all the waveforms are pre-processed through a normalization procedure (Section 2.3.1) before being used by the ANN model.

*2.2.1.1. Framingham Heart Study*. In the Framingham Heart Study (FHS) Original, Offspring, Third Generation Cohorts, each participant underwent arterial tonometry data collection [27, 43, 44]. In this work, we employ the uncalibrated carotid waveforms that were measured by applanation tonometry for a total number of $N$ = 6697 (53% women) participants in the study. The study protocol was approved by the Boston University Medical Campus and Boston Medical Center Institutional Review Board (IRB), and the study participants were consented prior to the study. The population/baseline characteristics are determined in terms of mean ± standard error of the mean (SEM) as 50.6 ± 0.2 years, 129.0 ± 0.2 mmHg, 68.1 ± 0.1 mmHg, 60.9 ± 0.2 mmHg, 62.2 ± 0.2 bpm, and 27.1 ± 0.1 kg/m$^2$ for age, brachial systolic BP, brachial diastolic BP, brachial pulse BP, heart rate (HR), and body mass index (BMI), respectively.

*2.2.1.2. HMRI iPhone Heart Study*. In the HMRI iPhone Heart Study [28], participants underwent carotid artery waveform recordings using three different device platforms: a commercial tonometry device (i.e., ATCOR Medical SphygmoCor [45]), an optical handheld device (called Vivio [42, 46]), and a smartphone camera (i.e., an iPhone [28]). The study protocol was approved by the Quorum Review IRB. A total number of 2312 uncalibrated carotid waveforms are selected from participants of the HMRI Heart Study for the database employed in this work. These waveforms are normalized and resampled (as described later in Section 2.3) for constructing the ML model. The population/baseline characteristics are determined in terms of mean ± SEM as 55.5 ± 2.2 years, 119.6 ± 1.7 mmHg, 75.9 ± 0.9 mmHg, 63.9 ± 1.2 bpm, and 25.6 ± 0.5 kg/m$^2$ for age, cuff systolic BP, cuff diastolic BP, HR, and BMI, respectively.

**2.2.2. Synthetic database.** A synthetic database is created using waveforms with exact IF frequencies and parameters (i.e., $\omega_1$, $\varphi_1$, $R_s$, $\omega_2$, $\varphi_2$, $R_d$, c) and following the below procedure:

1. Application of a normalized IF analysis (described later in Section 2.3.1) to all the carotid waveforms of both HMRI and FHS databases

2. Determination of the normalized IF parameters and their physiological ranges

3. Determination of two-dimensional regions for different pairs of the scaled IF parameters in order to distinguish the pairwise physiological regions and relationships

4. Generation of a uniform synthetic dataset using uniform meshing on the normalized IF parameters within the pairwise physiological regions. Normalized waveforms $\hat{P}(\tau)$ are generated by substituting the normalized IF parameters (described later in Section 2.3.1) into the reformulated version of the IF method (Eq (3)).

Such a synthetic database is employed in order to expand the parameter set and improve training of the IF method. Although the real-world clinical databases used here are tremendously valuable for developing ML models, they are not uniformly distributed within physiological ranges of interest. Indeed, the majority of the clinical waveforms in this work (Section 2.2.1) represent those IF parameters which tend to aggregate near the center of physiological regions. Therefore, the areas closest to the margins (which are more likely associated with cardiovascular diseases) are represented by fewer data points. As such, the synthetic database generated in this study, which can uniformly cover the whole physiological region of the normalized IF parameters (based on the previously reported range of physiological IF parameters), is ultimately needed to supplement the clinical database in order to achieve a more globally-accurate ML model. A fixed size of $N = 500$ datapoints is considered as the standard waveform size for the generated synthetic waveforms (see Section 2.3.2).

## 2.3. Data pre-processing

**2.3.1. Waveform normalization procedure.** The IF parameters are not dependent on the unit of the recorded signal; however, when it comes to the collection, archiving, or analysis of a substantial number of datapoints for the IF method (e.g., towards machine learning, deep learning, etc.), it is naturally highly effective to reduce the size of the archive via a normalization of the waveforms without loss of generality (which can save enormous storage and time especially in the big-data studies). Such a normalization is important to ensure that the developed model is sensor (or device) agnostic, especially since the heterogeneous clinical databases contain measurements taken by different apparatus (whose raw waveforms can be of arbitrary units). As such, a new standard coordinate system has been proposed [37] for arterial waveforms through which measurements of different devices (and even different species [37]) fall within the same range of signals and IF parameters, enabling the inclusion of any arbitrary raw arterial signal. In this spirit, a further normalization in time enables the mapping of all arterial waveforms onto the same length of the cardiac cycle (T′ = 1). The normalization procedure is as follows:

1. Subtract the minimum value $P_{min} = P(t)$ of the signal (given in any arbitrary measuring unit) from the measured $P(t)$ at all times of the entire cardiac cycle (i.e., $P(t) - P_{min}$, $0 \leq t \leq T$)

2. Divide the resulting waveform by its range over the entire cardiac cycle (i.e., $\hat{P}(t) = (P(t) - P_{min})/(P_{max} - P_{min}), 0 \leq t \leq T$)

3. Normalize in time by scaling $t$ with the length $T$ of the entire cardiac cycle (i.e., $\hat{P}(\tau) = \hat{P}(\tau(t)), \tau = t/T, 0 \leq \tau \leq 1$)

The procedure results in a scaled waveform $\hat{P}(\tau)$ corresponding to the original signal $P(t)$. The IF method can then be applied to the scaled waveform, thereby extracting new (non-dimensional) IF parameters as a result. Together with Eqs (1) through (4), the non-dimensional normalized IF parameters can be expressed in terms of the original (main) IF

parameters as follows:

$$\hat{\omega}_1 = \omega_1 T, \; \hat{\omega}_2 = \omega_2 T, \; \hat{c} = \frac{c - P_{min}}{P_{max} - P_{min}}, \; \hat{T}_0 = \frac{T_0}{T},$$

$$\hat{R}_s = \frac{R_s}{P_{max} - P_{min}}, \; \hat{R}_d = \frac{R_d}{P_{max} - P_{min}}, \; \hat{ER} = \frac{\hat{R}_s}{\hat{R}_d}, \; \hat{\varphi}_1 = \varphi_1, \hat{\varphi}_2 = \varphi_2. \quad (5)$$

The overall procedure yields a normalization of the range of waveform values as well as the length of the cardiac cycle. Hence this technique enables cross-platform comparisons of IF applied to any arterial waveform (measured by an arbitrary sensor platform), regardless of cardiac cycle length and initial measuring units. Furthermore, a model developed based on this normalization can be used for any species or any sensor platform with any arbitrary sampling rate.

**2.3.2. Waveform resampling procedure.** The ANN model proposed in this work (Section 2.4) is constructed so as to require a normalized dicrotic notch time and a scaled carotid waveform as the input; the outputs of the model are the correspondingly scaled IF parameters (i.e., $\hat{\omega}_1, \hat{\omega}_2, \hat{R}_s, \hat{\varphi}_1, \hat{c}$). In particular, the ANN model inputs the discrete data-points of a scaled carotid waveform. Since different measurement devices have different sampling rates, the cardiac cycle period is neither the same for different individuals nor for the same individual (e.g., beat to beat variations). Therefore, the normalized carotid waveforms are not of the same datapoint (vector) size. To resolve such sampling discrepancies towards building a global ANN model, a fixed size of $N = 500$ datapoints per cardiac cycle is considered as the standard waveform size that is input into the model. Hence, all considered waveforms are down/over-sampled before being employed in the model, thereby generating inputs of uniform dimension for the network. This facilitates consideration of signals from any measurement device with any arbitrary sampling rate (in addition to any arbitrary unit via the scaling procedure of Section 2.3.1). The waveform down/over-sampling process is performed by the use of spline interpolation to temporal discretization space of $\mathbb{R}^{500}$ (using the MATLAB Interp1 function).

## 2.4. Sequentially-reduced ANN model for solving the IF $L_2$ optimization

Since the $L_2$ optimization formulation originally introduced for IF calculations [20] is computationally expensive, we devise a sequentially-reduced ANN to effectively map the decoupling time (the notch time) and the waveforms to IF parameters (e.g., $\omega_1, \omega_2$). There are several network architectures that can be considered for this regression task [7], including FNNs, recurrent neural networks (RNNs), or temporal convolutional neural networks (TCNNs). The training of FNN is much more efficient and less prone to overfitting when compared with RNNs or TCNNs, although it requires inputs of a fixed size (unlike, e.g., RNNs). This, however, has been addressed by the proposed resampling of waveforms into uniform dimensions (Section 2.3.2).

As illustrated in Fig 2, an FNN structure consists of one input layer, $L$ hidden layers, and one output layer. The target FNN should map the high-dimensional input vector $\boldsymbol{x}$ into an output vector $\boldsymbol{y}$ of a significantly lower dimension (size). This results in a network of a sequentially-reduced structure, with uniformly decaying numbers of neurons (i.e., $W_0 \succ W_1 \succ W_2 \succ \ldots \ldots \succ W_L \succ W_{L+1}$ for a width $W_l$ corresponding to layer $l$ for $0 \le l \le L+1$). In our implementation, the number of neurons is reduced by half in each hidden layer (i.e., $W_l = W_1/2^{l-1}$ for $1 \le l \le L$). For such an FNN structure, the forward propagation of the network is

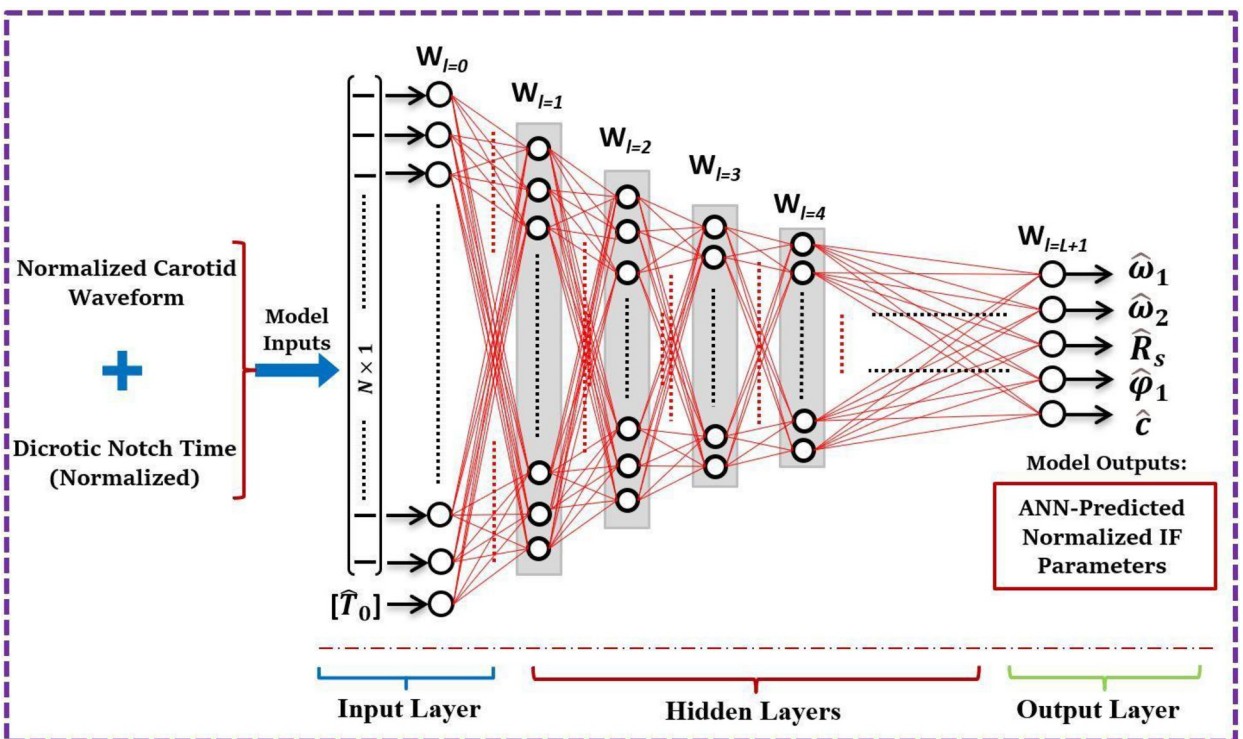

**Fig 2. Structural schematic of the sequentially–reduced feedforward ANN model for predicting the IF method outputs from a single carotid waveform with a given dicrotic notch time (the decoupling time).** Here, $L$ is the number of hidden layers.

given by:

$$\boldsymbol{y}^{(0)} = \boldsymbol{x},$$

$$\boldsymbol{y}^{(l)} = f_{act}(w^{(l)}\boldsymbol{y}^{(l-1)} + \boldsymbol{b}^{(l)}), l = 1, 2, \ldots, L,$$

$$\boldsymbol{y} = \boldsymbol{y}^{(L+1)} = w^{(L+1)}\boldsymbol{y}^{(L)} + \boldsymbol{b}^{(L+1)}, \tag{6}$$

where $w^{(l)}$ are the weight matrices, $\boldsymbol{b}^{(l)}$ are the bias vectors, and $f_{act}$ is the nonlinear activation function. In this work, we use the Swish activation function $f_{act}(x) = x*sigmoid(x)$ [47].

The inputs of the network are the normalized notch time and the 500-point interpolated waveform, leading to a space of dimensionality $\mathbb{R}^{501}$. The outputs of the model are the scaled IF parameters $\hat{\omega}_1, \hat{\omega}_2, \hat{R}_s, \hat{\varphi}_1$ and $\hat{c}$. Other IF parameters can be analytically computed from these five outputs (using the continuity and periodicity constraints) as

$$\hat{\varphi}_2 = \tan^{-1}\left(\frac{\sin(\hat{\varphi}_1)\sin(\hat{\omega}_2\hat{T}_0) - \sin(\hat{\omega}_1\hat{T}_0 + \hat{\varphi}_1)\sin(\hat{\omega}_2)}{\sin(\hat{\omega}_1\hat{T}_0 + \hat{\varphi}_1)\cos(\hat{\omega}_2) - \sin(\hat{\varphi}_1)\cos(\hat{\omega}_2\hat{T}_0)}\right) \text{ and}$$

$$\hat{R}_d = \frac{\hat{R}_s\sin(\hat{\varphi}_1)}{\sin(\hat{\omega}_2 + \hat{\varphi}_2)}. \tag{7}$$

Since the output variables have different scales (ranges), it is necessary to perform feature scaling [48] in order to train a network to have uniform accuracy across all output variables. The feature scaling for a variable $y$ is given by $\hat{y} = (y - y_-)/\sigma_y$, where $y_-$ and $\sigma_y$ are the mean

**Table 1. Values of the training hyper–parameters employed in this work.**

| Hyper-parameter | Notation | Range |
|---|---|---|
| Number of hidden layers | $L$ | 3, 4, 5 |
| Number of neurons of the first hidden layer | $W_1$ | 128, 256, 512 |
| Regularization type | $Reg$ | $L_1, L_2$ |
| Regularization coefficient | $\lambda$ | $10^{-5}, 10^{-6}, 10^{-7}$ |

and standard deviation of $y$, respectively. During the training, the weights and biases of the network are adjusted by minimizing a loss function. The loss function employed here is the mean squared error (MSE). An $L_1$ (resp. $L_2$) weight regularization term, which is the sum of absolute (resp. squared) values of the weights, is added to the loss function to avoid overfitting. The amount of regularization is controlled by using a hyper-parameter $\lambda$. The optimal weights and biases are obtained by employing the Adam stochastic optimizer [49]. In each epoch, the training data set is shuffled and then divided into several mini-batches. The weights and biases are updated every time the loss function is minimized on the mini-batch. The network parameters converge after the training is performed on a sufficiently large number of epochs. The convergence speed of the training is controlled by the learning rate (default value of $10^{-3}$). Each network is trained with 10 restarts in order to avoid the influence on training of the random initialization of weights and biases [50].

The hyper-parameters employed in the network training of this work are listed in Table 1. A grid-search of the hyper-parameters is performed to find the optimal network configuration [7]. Training is implemented in Keras with Tensorflow as the backend [51]. The trained network with the smallest validation error is ultimately selected for our model.

## 2.5. Training data size and sensitivity analysis

We employ a total number of N = 6000 clinical waveforms (i.e., a combination of the entire HMRI database and 55.1% of the FHS database) as well as N = 8208 synthetically generated waveforms to design (i.e., train, validate, and generalize) our proposed FNN model. For the training data, 80% of the clinical waveforms (N = 4800) and 100% of the synthetic waveforms (N = 8208) are used. Of the remaining clinical waveforms, 10% (N = 600) are employed as the validation data for the model selection, and the final 10% (N = 600) are reserved for the test data in the estimation of generalization accuracy. The rest of the clinical data (remaining 44.9% of FHS) is used for a blind clinical test.

Sensitivity analysis is performed to assure that the training data size is sufficient. To this end, the training data size is decreased gradually from a total training data size corresponding to N = 13,008 cases (i.e., 4,800 clinical waveforms and 8,208 synthetic waveforms). The mean squared error (MSE) for both training and validation losses is correspondingly measured. We use the same validation population (N = 600 clinical waveforms) to compare the accuracy of the trained models.

As described previously, our clinical pressure waveform database is provided by measurements from different devices subject to different resolutions. Since the input size of the pressure waveform is set to n = 500 in our proposed ANN model, all waveforms are down/oversampled to n = 500 (as described in Section 2.3.2) prior to being fed into the model. The effects of this resampling on our model's predictions are studied in Section 3.4.

## 2.6. Analysis and statistical methods

Fig 3 presents a flowchart diagram summarizing both the (standard) $L_2$ optimization-based approach and the proposed ML-based approach for computation of the IF parameters. The

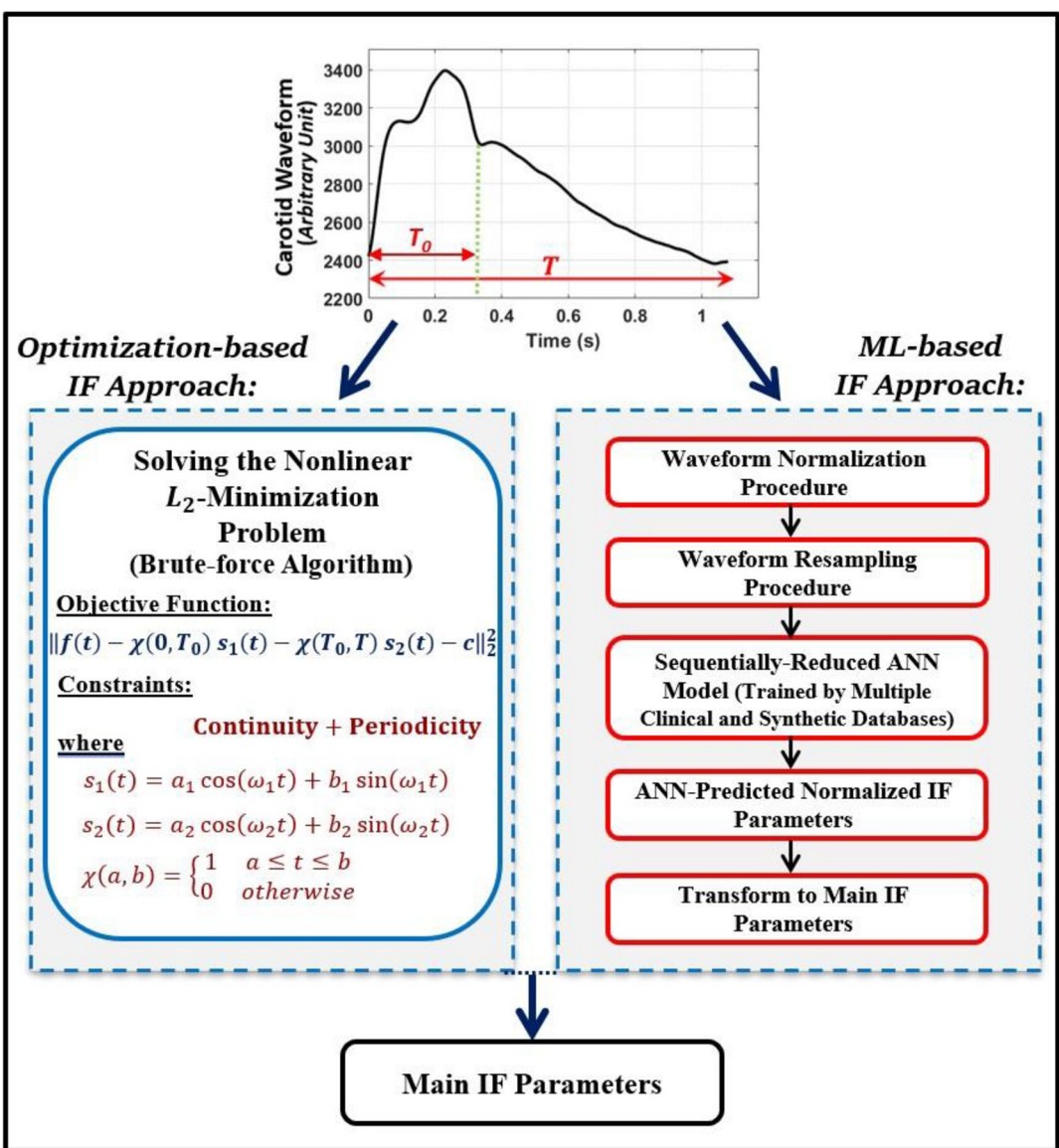

**Fig 3. Flowchart diagram summarizing the standard optimization–based IF approach and the proposed ML–based IF approach.**

agreement level, bias, and precision between the ANN-based IF predictions and the $L_2$ optimization-based IF parameters (presented in Section 3) are assessed by employing different metrics: a regression (Pearson correlation coefficients), root mean square errors (RMSE), relative errors of the ensembles, point-wise average relative errors, Bland-Altman analysis [52, 53], and a histogram analysis of the absolute errors.

## 2.7. Blind clinical test

We further test the developed model using the stratified blind test technique. The model is blindly tested on N = 3009 additional Framingham Heart Study (FHS) clinical waveforms that are set aside prior to our ML model development (a so-called "external validation").

**Table 2. Generalization accuracy results for the optimal ANN model.**

| Output | Range | RMSE (ML-based Vs. Optimization-based Approach) | Relative Error of the Ensemble (ML-based Vs. Optimization-based Approach) |
|---|---|---|---|
| $\hat{\omega}_1$ | [60.4, 143.5] | 0.63 | 0.71% |
| $\hat{\omega}_2$ | [26.9, 152.3] | 1.49 | 2.44% |
| $\hat{R}_s$ | [0.49, 0.94] | 0.006 | 0.84% |
| $\hat{\varphi}_1$ | [-1.06, 0.16] | 0.014 | 4.74% |
| $\hat{c}$ | [0.14, 0.51] | 0.005 | 1.55% |

# 3. Results

## 3.1. Training

The optimal network obtained in this work contains four hidden layers ($L = 4$) of widths $W$ = 256, 128, 64, and 32 neurons. The training performed with the $L_2$ loss function employs a value of $10^{-6}$ as the coefficient of the regularization. The generalization test error of the final model is presented in Table 2.

## 3.2. Sensitivity of the network design

The sensitivity of the accuracy to the training data size is shown in Fig 4. MSE loss decreases gradually for both training loss and validation loss after increasing the relative training size from 20% to 100% of the training data (Fig 4). The 100% of the training data corresponds to N = 13,008 waveforms. It is noteworthy that MSE is in absolute units corresponding to its calculation from the normalized outputs.

## 3.3. Blind clinical test

Here, we use the blind clinical test to assess the efficacy of our proposed ML-based IF model with different indices and figures. The resulting RMSE and relative errors of the blind clinical test (described in Section 2.7) are presented in Table 3. To investigate and confirm the agreement between the FNN-based IF parameter predictions and the standard L2-based IF calculations, we also used blind test data and plotted different statistical analysis figures (i.e.,

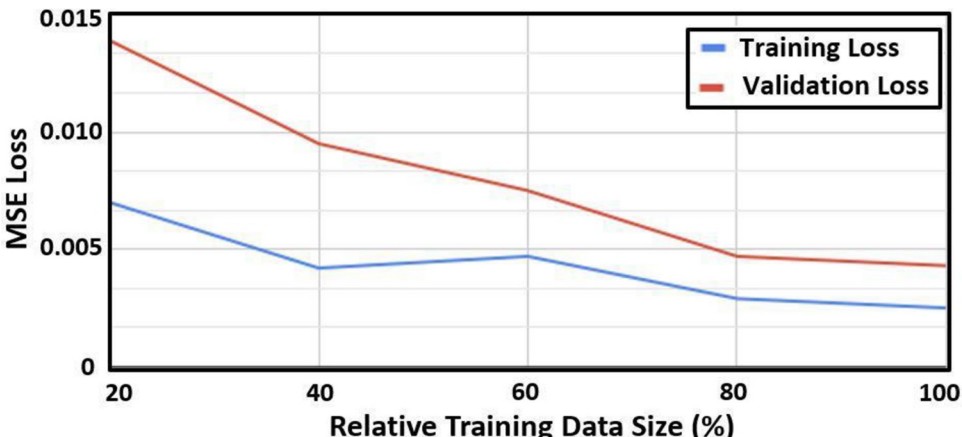

**Fig 4. Sensitivity of precision in terms of mean squared error (MSE) loss (for training and validation) versus the relative training data size.** Here, 100% of the training size corresponds to N = 13,008 waveforms. MSE is in absolute units corresponding to its calculation from the normalized outputs.

**Table 3. Errors and ranges of the blind clinical test results for proposed ANN design (N = 3009).**

| Output | Range | RMSE (ML-based Vs. Optimization-based Approach) | Relative Error of the Ensemble (ML-based Vs. Optimization-based Approach) |
|---|---|---|---|
| $\hat{\omega}_1$ | [75.0, 155.2] | 1.81 | 1.82% |
| $\hat{\omega}_2$ | [19.2, 71.3] | 2.70 | 5.62% |
| $\hat{R}_s$ | [0.43, 0.89] | 0.0139 | 2.01% |
| $\hat{\varphi}_1$ | [-1.269, -0.008] | 0.0349 | 9.99% |
| $\hat{c}$ | [0.18, 0.58] | 0.0144 | 4.00% |

regression plots, Bland-Altman plots, and error histograms) for the scaled IF parameters (Figs 5 and 6). The diagonal dotted line in the regression plots represents equality. In the Bland-Altman plots, limits of agreement (LoA) are shown by the two horizontal solid lines in purple. The Bland-Altman analysis helps with confirming a low bias for the ML-based IF parameter predictions. The error histograms also help with investigating and confirming low occurrence of high errors for the ML-based IF parameter predictions.

## 3.4. Effects of measurement device and sampling rate

As mentioned earlier, the ML model requires that every waveform undergoes a normalization and a resampling to 500 samples (datapoints) per cardiac cycle. However, the model dependency on the original sampling rate of the waveform or the measurement device should be assessed to ensure that the model stays accurate across different sampling rates and measurement devices. The effects of measurement device and signal sampling rate on IF parameter predictions produced by the model are shown in the scatter and box-whisker plots of Figs 7

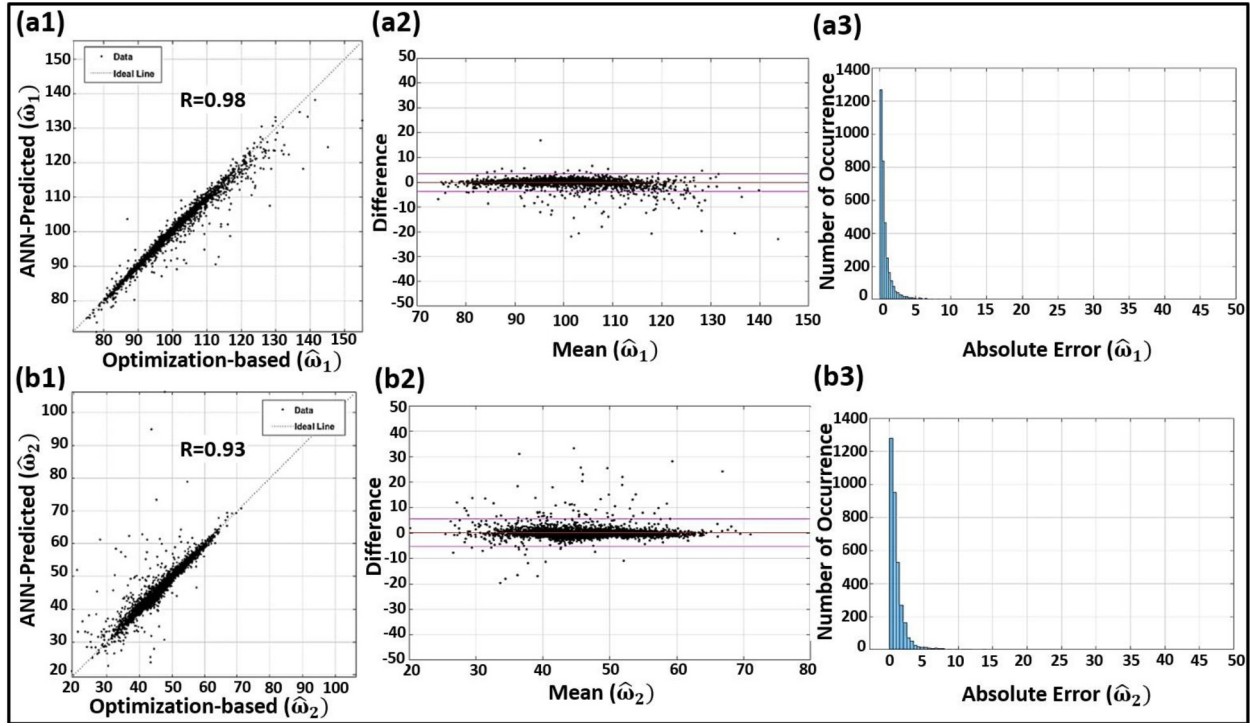

**Fig 5.** Statistical analysis of the blind clinical tests in terms of regression plots (left column), Bland–Altman plots (middle column), and error histograms (right column) of the scaled IFs $\hat{\omega}_1$ (top row: **a1**, **a2**, and **a3**, respectively) and $\hat{\omega}_2$ (bottom row: **b1**, **b2**, and **b3**, respectively).

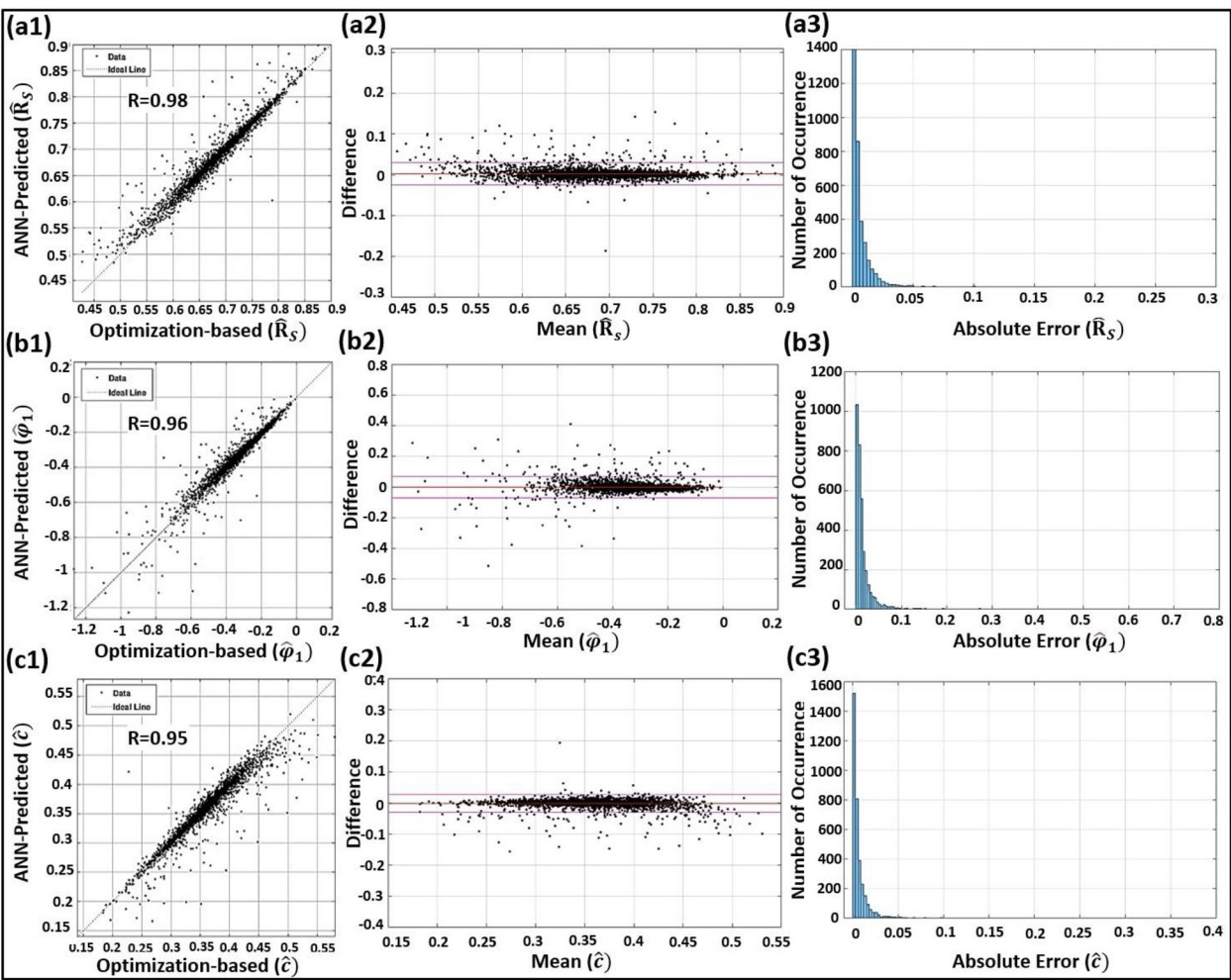

**Fig 6.** Statistical analysis of the blind clinical tests in terms of regression plots (left column), Bland–Altman plots (middle column), and error histograms (right column) of the scaled first intrinsic envelope ($\hat{R}_s$; top row: **a1**, **a2**, and **a3**, respectively), the scaled first intrinsic phase ($\hat{\varphi}_1$; middle row: **b1**, **b2**, and **b3**, respectively), and the scaled fitting constant ($\hat{c}$; bottom row: **c1**, **c2**, and **c3**, respectively).

and 8 where errors are presented for $\hat{\omega}_1$, $\hat{\omega}_2$, $\hat{R}_s$, $\hat{\varphi}_1$, and $\hat{c}$ computed from the ANN model using all the clinical data that are used in the design process. The vector size of these waveforms ranges from 50 to 169 (for iPhone measurements), 150 to 619 (for Tonometry measurements), and 558 to 1560 (for Vivio measurements). The box-whisker plot here displays a five-index summary of error for each device by showing the minimum, first quartile, median, third quartile, and maximum.

## 4. Discussion

An ML framework provides a viable alternative to the mechanistic models in health monitoring and disease diagnosis. However, model objectives should be designed carefully in order to achieve robust and successful ML implementations that can extract crucial physiological information. In order to avoid high-dimensional classification settings arising from diagnosing cardiovascular diseases, we have posed the problem of such clinically-relevant analysis in terms of the intrinsic frequency (IF) domain. In particular, we have introduced a sequentially-reduced

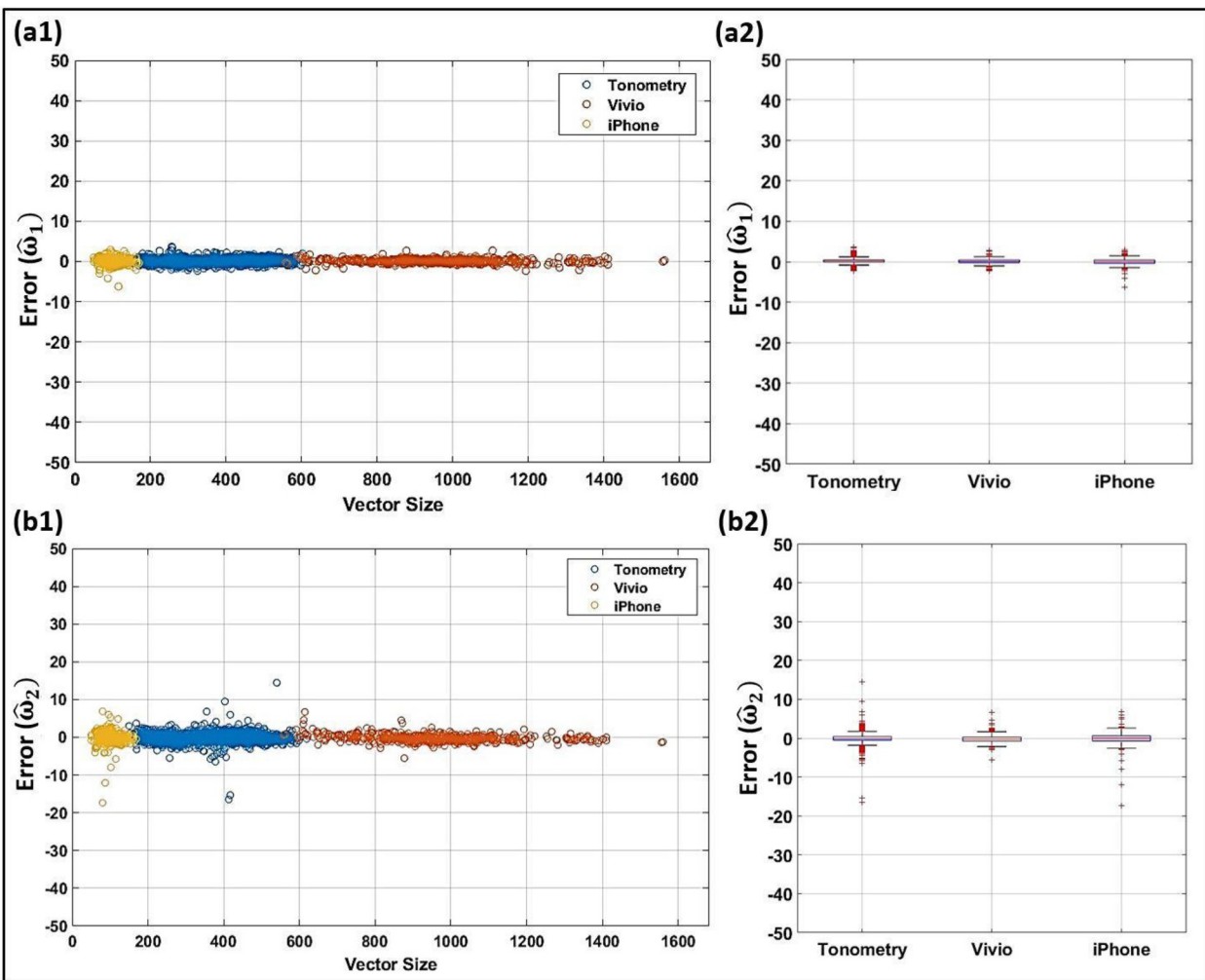

**Fig 7. Effects of the original sampling rate and measurement device on predictions of IFs from the ANN model.** Scatter plots and box–whisker plots of the error for the scaled IFs $\hat{\omega}_1$ (top row: **a1** and **a2**, respectively) and $\hat{\omega}_2$ (bottom row: **b1** and **b2**, respectively). The measurement devices include Tonometry (shown by blue circles), Vivio (shown by red circles), and iPhone (shown by orange circles).

FNN model which directly extracts IFs of arterial pressure waveforms. The model has been designed (i.e., trained, validated and tested) using a heterogeneous database that includes both clinical pressure waveforms as well as synthetic waveforms. The clinical dataset employed has been generated from two different clinical study cohorts (FHS and HMRI) with carotid waveforms measured using three different device (sensor) platforms: traditional tonometry devices (with piezoelectric sensors), an optical wireless tonometer (Vivio), and an iPhone camera. We have performed an external blind test on the model using 3009 clinical waveforms, where our presented results demonstrate excellent agreement between the FNN-based IF and the standard non-convex optimization formulation (i.e., the brute-force $L_2$ method).

The FNN-based IF model proposed in this work has been constructed using four sequentially-reduced hidden layers with 256, 128, 64, and 32 neurons at the first, second, third, and fourth layer, respectively. A total of 13008 waveforms (N = 4800 clinical waveforms and N = 8208 synthetic waveforms) have been used to train, validate, and test the designed model. Each waveform goes through a pre-processing step (Section 2.3) that includes normalization and resampling (500 timesteps per cycle). These pre-processed waveforms are used as input

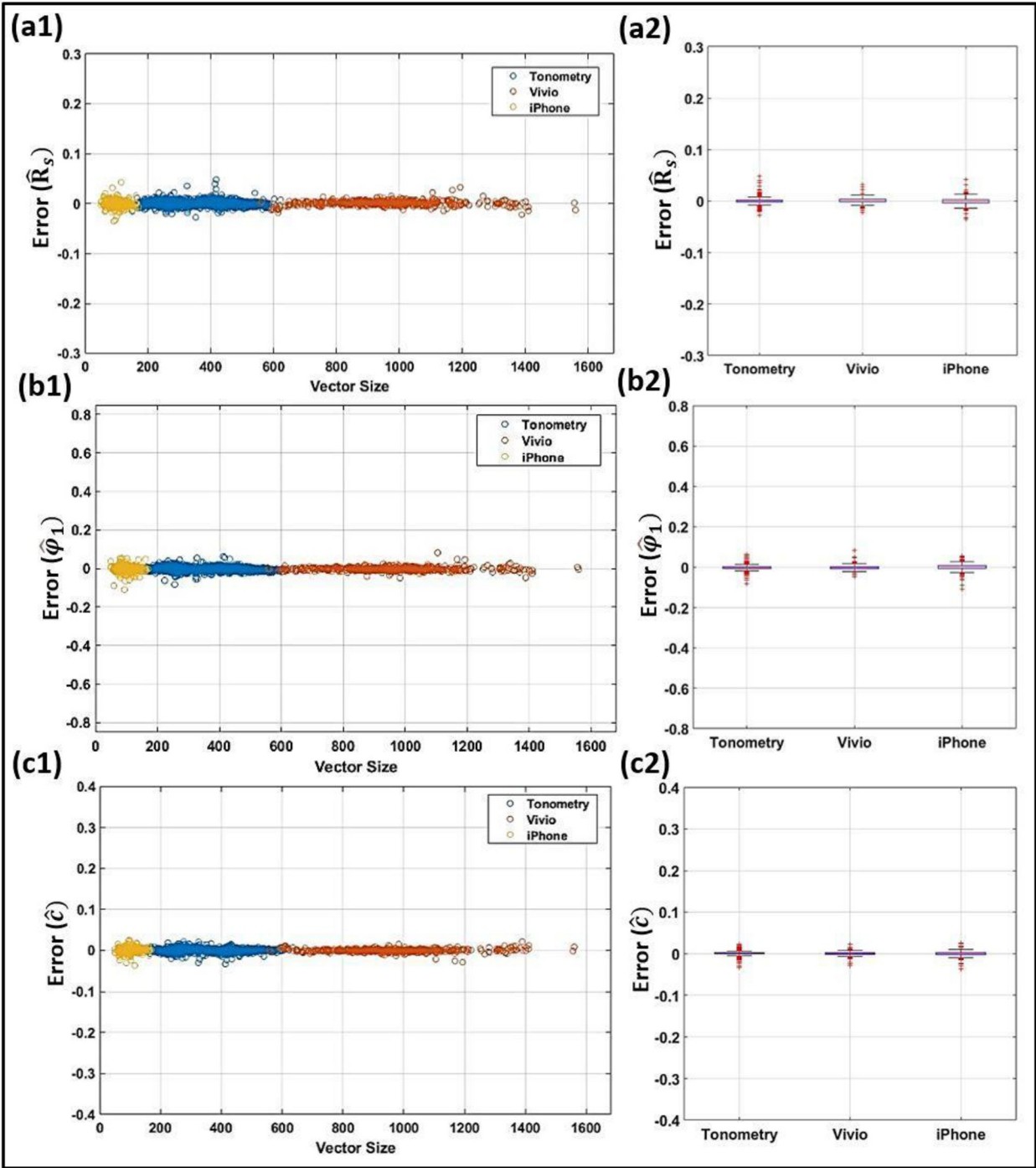

**Fig 8. Effects of the original sampling rate and measurement device on predictions of related IF parameters from the ANN model.** Scatter plots and box–whisker plots for the scaled first intrinsic envelope ($\hat{R}_s$; top row: **a1** and **a2**, respectively), the scaled first intrinsic phase ($\hat{\varphi}_1$; middle row: **b1** and **b2**, respectively), and the scaled fitting constant ($\hat{c}$; bottom row: **c1** and **c2**, respectively). The measurement devices include Tonometry (shown by blue circles), Vivio (shown by red circles), and iPhone (shown by orange circles).

for the FNN model. The corresponding outputs of the network are the two scaled intrinsic frequencies $\hat{\omega}_1, \hat{\omega}_2$ and associated envelope and phase parameters $\hat{R}_s, \hat{\varphi}_1, \hat{c}$ (other IF parameters can be computed analytically from these five outputs, as described in Section 2.4). These scaled

IFs can get translated back to the unscaled (dimensional) IF parameter values for each individual waveform using Eq (5). As shown in Table 2, $\hat{\omega}_1$ computed from the designed FNN-based IF model results in an RMSE of 0.6 (over a range of 60 to 143) with a relative error of 0.71%. The corresponding relative error for $\hat{\omega}_2$ was 2.4% with an RMSE = 1.49 (over a range of 27 to 152). The sensitivity to the training data size on the accuracy attained by the proposed model has been tested by gradually increasing the relative size of the training data (from 20% to 100% of the data). As shown in Fig 4, additional data after 80% of the relative size has negligible effect on the improvement of the network accuracy (<0.001).

We have also performed an external (stratified) blind test on the model using a random assortment of clinical pressure waveforms from FHS clinical database that were set aside before constructing and training the model. The resulting correlation coefficients in this blind test were R = 0.98 for the first IF ($\hat{\omega}_1$) and R = 0.93 for the second IF ($\hat{\omega}_2$). The correlation coefficients are higher than R = 0.95 for all other IF-related parameters (other outputs). Such high correlations attest to the efficacy of our proposed ML-based IF model. As shown in Table 3, the intrinsic frequencies $\hat{\omega}_1$ and $\hat{\omega}_2$ demonstrate low relative errors of 1.82% (RMSE = 1.81; range: 75–155) and 5.62% (RMSE = 2.70; range: 19–71), respectively. The Bland-Altman plots of Figs 5 and 6 additionally confirm the excellent agreement between the FNN-based IF parameter predictions and the standard $L_2$-based IF calculations.

Our models only require that every waveform undergoes a normalization and a resampling to n = 500 samples (datapoints) per cardiac cycle. Figs 7 and 8 assess the performance of the model as a function of different sampling rates and measurement device platforms. As demonstrated in such scatter and box-whisker plots, the error is independent of the sensor platform and sampling rate. Hence, any arterial waveform acquired from any arbitrary measurement device with any arbitrary sampling rate can be used as an input to the FNN-based IF model. Although the model performed very well for majority of waveforms with low original vector size, over-sampling of the waveforms during preprocessing can cause error as most of the relatively high errors occurred for the waveforms with lower original vector sizes. In addition to its accuracy and independence from chosen measurement apparatus, the devised FNN algorithm is computationally efficient (which is a particular motivation of the present work). The online predictions are almost real-time for each resampled waveform in blind clinical tests (3.94e-5 seconds on a laptop with a 2.3 GHz Quad-Core Intel Core i5 CPU). The offline training takes around 14 minutes per network (100% training dataset; maximum 5 layers; 512, 256, 128, 64 and 32 possible neurons; one restart). In contrast, the L2 optimization for IF computations takes multiple seconds per waveform (around 9 to 40 seconds depending on the optimization settings via the same laptop).

In this manuscript, we have presented an FNN for determining IF parameters of a coupled system with only one decoupling point (the time of the dicrotic notch, i.e., when the aorta and the heart decouple). The methodology can be easily extended to coupled systems with multiple (two or more) decoupling times in more physically complex systems. As is the case for only two IFs, extracting more IFs from such systems is computationally expensive when a brute-force $L_2$ minimization technique is used. However, the computational times demonstrated for the FNN-based IF-model of this work imply that an ML-type model may also be able to extract multiple IFs of interest within a fraction of a second.

## 4.1. Limitations and future work

The main limitation for the current study is that the clinical value of the IF parameters predicted by our proposed FNN-based approach (for diagnostic and prognostic cardiovascular diseases) has not been assessed. This will be studied in our future work. Furthermore. the

methodology proposed in this work will be extended and applied to other complex systems with multiple (two or more) decoupling times.

## 5. Conclusions

In this work, a sequentially-reduced FNN model has been proposed for the prediction of IF parameters without the need to directly solve the standard IF non-convex optimization problem (and corresponding $L_2$ minimization). The model inputs a normalized carotid waveform and produces IFs and envelope/phase parameters as the output. The FNN-based IF parameters demonstrate excellent agreement with the $L_2$-based IF parameters in an external blind test on the clinical data. Since a normalized waveform is used as the model input, any carotid waveform recorded by any sensor platform and from any species (e.g., rat or rabbit) can be accepted by the model. Hence the model is compatible with any device that collects invasive or non-invasive waveforms.

## Acknowledgments

This manuscript was not prepared in collaboration with investigators of the Framingham Heart Study and does not necessarily reflect the opinions or conclusions of the Framingham Heart Study or the NHLBI. The authors would like to thank Dr. Faisal Amlani for productive discussion.

## Author Contributions

**Conceptualization:** Rashid Alavi, Hossein Gorji, Niema M. Pahlevan.

**Formal analysis:** Rashid Alavi, Qian Wang.

**Investigation:** Rashid Alavi, Qian Wang.

**Methodology:** Rashid Alavi, Qian Wang, Hossein Gorji.

**Project administration:** Niema M. Pahlevan.

**Supervision:** Hossein Gorji, Niema M. Pahlevan.

**Validation:** Rashid Alavi, Qian Wang, Niema M. Pahlevan.

**Visualization:** Rashid Alavi, Qian Wang.

**Writing – original draft:** Rashid Alavi, Qian Wang.

**Writing – review & editing:** Hossein Gorji, Niema M. Pahlevan.

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
