## [Decision Letter · Decision Letter 0]

9 Mar 2023

PONE-D-23-02524A Machine Learning Approach for Computation of Cardiovascular Intrinsic FrequenciesPLOS ONE

Dear Dr. Niema M. Pahlevan,

Thank you for submitting your manuscript to PLOS ONE. After careful consideration, we feel that it has merit but does not fully meet PLOS ONE’s publication criteria as it currently stands. Therefore, we invite you to submit a revised version of the manuscript that addresses the points raised during the review process.

We look forward to receiving your revised manuscript.

Kind regards,

Yih Miin Liew

Academic Editor

PLOS ONE

Journal Requirements:

"Niema M. Pahlevan holds equity in Avicena LLC and has a consulting agreement with Avicena LLC. The remaining authors do not declare any competing interest."

**Comments to the Author**

1. Is the manuscript technically sound, and do the data support the conclusions?

Reviewer #1: Yes

Reviewer #2: Yes

2. Has the statistical analysis been performed appropriately and rigorously? 

Reviewer #1: Yes

Reviewer #2: Yes

3. Have the authors made all data underlying the findings in their manuscript fully available?

Reviewer #1: Yes

Reviewer #2: Yes

4. Is the manuscript presented in an intelligible fashion and written in standard English?

Reviewer #1: Yes

Reviewer #2: Yes

5. Review Comments to the Author

Reviewer #1: This paper attempts to use machine learning approach to replace the optimization approach to compute the cardiovascular intrinsic features.

A. The abstract is well written.

B. The introduction is clear.

C. The methodology is clear.

D. Results:

1. Please show the computational time required for the optimization method as compared to the machine learning approach.

2. A lot of relevant and interesting figures are shown in the results section. It would be good if the author highlight the key information from the figures.

E. Discussion:

1. Although the Machine learning approach in principle shows comparable performance to the optimization approach, it would be interesting if the author can explain in what condition the system will perform badly as some of the data shows very poor performance.

Reviewer #2: This paper proposes a machine learning (ML) approach based on sequentially-reduced FNN model for the prediction of Cardiovascular Intrinsic Frequencies (IF) parameters. Experiment was conducted using heterogeneous datasets (2 clinical datasets, HRMI and FHS and synthetic database) and The results produced by the proposed model were benchmarked against the results from standard L2 optimization-based method to prove there is strong correlation of between IF parameters computed by ML model and parameters computed by L2 optimization-based method. Besides, the proposed FNN model can reduce the computation time as compared with the traditional approach. The methods used to design, develop train, validate, test, validate and blind clinical test the proposed model are clearly explained in this paper.

Comments:

1. Section 1 and Section 2: Description of Intrinsic Frequency (IF) method described in Section 1 and Section 2 has some repetitions and can be combined and focus in one of the section only. Introduction can give a brief description of this method.

2. Section 3 Results: Brief descriptions were provided for the results. The authors should elaborate more about the results presented in each table and figure. Beside using Figure, it will be good to have a table to show the comparison of results computed by the FNN model and L2 Optimization-based method.

3. Section 3 Results and Section 4 Discussion: Besides correlate your model to the L2 optimization methods, the authors should also explain the performance of the proposed model based on the evaluation metrics. For example, RMSE for some IF parameters is more than 1. Generally, the lower the value of the RMSE, the better the model is. RMSE more than 1 means that a model couldn't find a solution or (optimized very well).

4. Section 4 Discussions / Section 5 Conclusions: The main limitations found during the test/experiment should be discussed and any possible solution that could be applied to address these limitations.

5. Figure 7 and Figure 8 - the table legend used to indicate the color for each device is not very clear, it is hard to know what color is used in the diagram to represent each device. The authors may need to use word to explain what color is used to represent a device.

6. PLOS authors have the option to publish the peer review history of their article (what does this mean?). If published, this will include your full peer review and any attached files.

Reviewer #1: No

Reviewer #2: No

---

## [Author Response · Author response to Decision Letter 0]

29 Mar 2023

Responses to reviewers' comment are provided in the uploaded file.

---

## [Decision Letter · Decision Letter 1]

18 Apr 2023

A Machine Learning Approach for Computation of Cardiovascular Intrinsic Frequencies

PONE-D-23-02524R1

Dear Dr. Pahlevan,

We’re pleased to inform you that your manuscript has been judged scientifically suitable for publication and will be formally accepted for publication once it meets all outstanding technical requirements.

Kind regards,

Yih Miin Liew

Academic Editor

PLOS ONE

Additional Editor Comments (optional):

Reviewers' comments:

Reviewer's Responses to Questions

**Comments to the Author**

1. If the authors have adequately addressed your comments raised in a previous round of review and you feel that this manuscript is now acceptable for publication, you may indicate that here to bypass the “Comments to the Author” section, enter your conflict of interest statement in the “Confidential to Editor” section, and submit your "Accept" recommendation.

Reviewer #1: All comments have been addressed

Reviewer #2: All comments have been addressed

2. Is the manuscript technically sound, and do the data support the conclusions?

Reviewer #1: Yes

Reviewer #2: Yes

3. Has the statistical analysis been performed appropriately and rigorously? 

Reviewer #1: Yes

Reviewer #2: Yes

4. Have the authors made all data underlying the findings in their manuscript fully available?

Reviewer #1: Yes

Reviewer #2: Yes

5. Is the manuscript presented in an intelligible fashion and written in standard English?

Reviewer #1: Yes

Reviewer #2: Yes

6. Review Comments to the Author

Reviewer #1: The authors have addressed all the reviewer comments.

Reviewer #2: (No Response)

7. PLOS authors have the option to publish the peer review history of their article (what does this mean?). If published, this will include your full peer review and any attached files.

Reviewer #1: No

Reviewer #2: No

---

## [Editor Report · Acceptance letter]

27 Apr 2023

PONE-D-23-02524R1 

A Machine Learning Approach for Computation of Cardiovascular Intrinsic Frequencies 

Dear Dr. Pahlevan:

I'm pleased to inform you that your manuscript has been deemed suitable for publication in PLOS ONE. Congratulations! Your manuscript is now with our production department. 

Kind regards, 

on behalf of

Assoc Prof Ir Dr Yih Miin Liew 

Academic Editor

PLOS ONE